# BioAgent Bench: An AI Agent Evaluation Suite for Bioinformatics

**Dionizije Fa**[* 1] **Marko Čuljak**[* 2] **Bruno Pandža**[1] **Mateo Čupić**[1]

## Abstract

We introduce BioAgent Bench, an evaluation suite designed for measuring the performance and robustness of AI agents in common bioinformatics tasks. The suite consists of manually curated end-to-end tasks (e.g., RNA-seq, variant calling, metagenomics) accompanied by task-specific prompts and concrete output artifacts to support automated assessment. We evaluate frontier closed- and open-weight models across multiple agent harnesses, and use an LLM-based grader to score pipeline progress and outcome validity. We find that agents based on frontier LLMs can complete multi-step bioinformatics pipelines without elaborate custom scaffolding, often producing the requested final artifacts reliably. However, robustness tests reveal failure modes under controlled perturbations (corrupted inputs, decoy files, and prompt bloat), indicating that correct high-level pipeline construction does not guarantee reliable step-level reasoning. Finally, bioinformatics workflows often involve sensitive patient data or unpublished intellectual property, thereby making the use of cost-effective yet reliable local agents an imperative. Therefore, by releasing the code and the complementary resources comprising our suite, we aim to accelerate the development of such privacy-preserving agents.

github/bioagent-bench
github/bioagent-experiments

## 1. Introduction

Coding agents based on large language models (LLMs) have become a central component of developing computational workflows, including those in the life sciences. Bioinformatics is a particularly compelling domain for such agentic

systems, as many routine analyses involve chaining together command-line tools, managing heterogeneous file formats, and interpreting intermediate outputs in a structured and often highly subfield-specific way. Their complexity and a wide variety of plausible outputs complicate rigorous evaluation of such analyses.

Further, parameter and algorithm choices at early stages of a pipeline can drastically influence final results, thereby confounding ablation studies and weakening the robustness of drawn conclusions. These limitations make analysis steps difficult or impossible to label with strict pass/fail criteria without information loss. However, existing evaluations tend to collapse rich workflows into simplified question-answering or code-generation problems that fail to produce nuanced and actionable insights into an agent's behavior and failure modes.

At the same time, bioinformatics pipelines often operate on highly sensitive data, such as patient tumor sequencing samples or clinical metadata, where both privacy and intellectual property constraints prevent sharing raw datasets with third-party model providers or releasing them publicly (Zhou et al., 2024). In these settings, automating such pipelines requires models that can be run entirely within the institution's secure environment.

To address these gaps, we introduce a benchmark dataset of common bioinformatics tasks specifically designed to evaluate large language model (LLM) agents in realistic workflows that require tool use. The benchmark covers a diverse set of tasks that reflect everyday usage scenarios for bioinformatics practitioners. We utilize the benchmark to evaluate the performance of frontier closed- and open-weight models when used as agents in bioinformatics workflows. We show that frontier proprietary models can successfully solve the majority of benchmark tasks without extensive custom scaffolding, whereas the best-performing open-weight models underperform significantly.

However, we observe significant performance drops under controlled perturbations and inconsistencies across multiple runs. These results indicate that while present LLM technology is already capable of supporting and automating many routine bioinformatics workflows, additional work is needed to ensure they meet the high robustness, reliability, and safety standards of the field.

---

[*]Equal contribution [1]Entropic [2]TakeLab @ FER, University of Zagreb. Correspondence to: Dionizije Fa <entropic, dionizije.fa@outlook.com>.

*Proceedings of the $43^{rd}$ International Conference on Machine Learning*, Seoul, South Korea. PMLR 306, 2026. Copyright 2026 by the author(s).

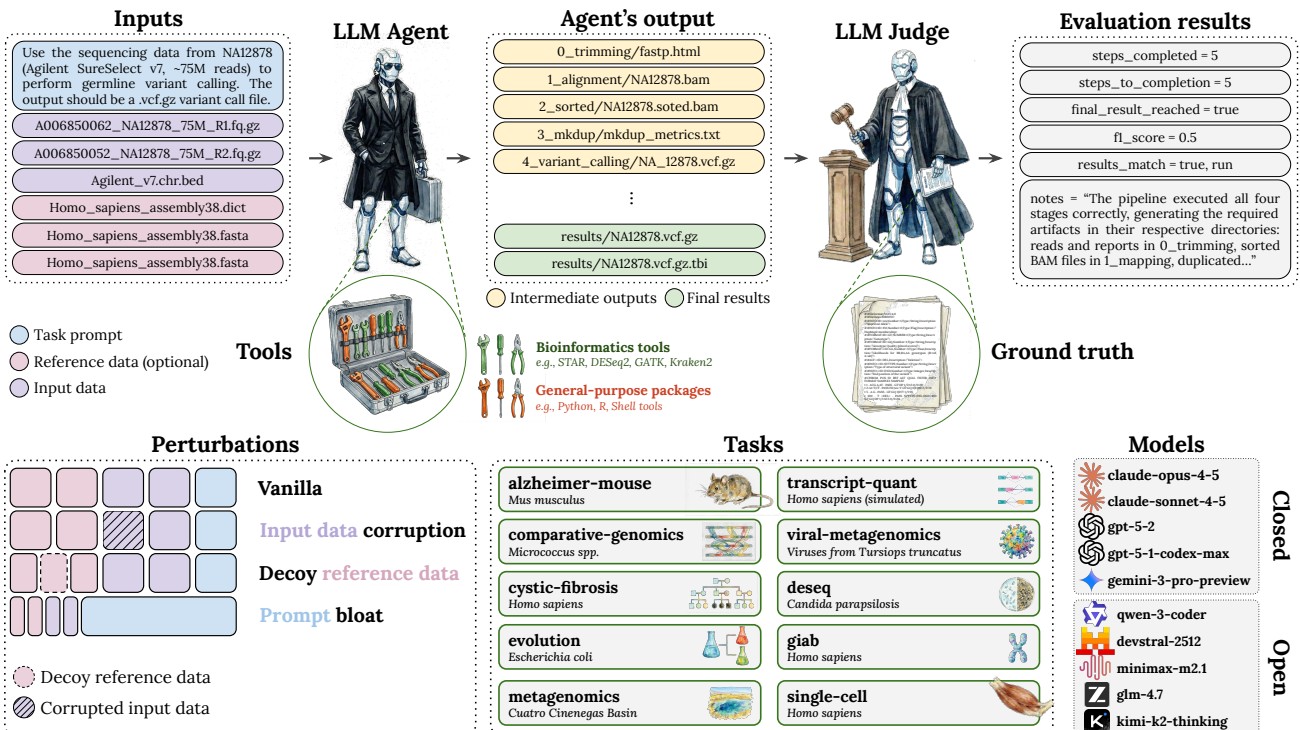

*Figure 1.* An overview of BioAgent Bench. Inputs to LLM agents consist of a task prompt, input data, and reference data. While solving the provided task, an agent can use general-purpose packages or specialized bioinformatics tools. After the agent finishes generation, LLM judge compares its outputs against the ground truth and produces evaluation results. In addition to the standard "vanilla" inputs, we also experiment with different perturbations to stress-test the agents. We focus our evaluation on 10 bioinformatics tasks (each designed around a different organism, virus, or entire ecosystem) and 10 models (5 open-weight and 5 closed-weight).

In summary, we contribute the following: (i) a benchmark dataset for evaluating AI agents on practical bioinformatics tasks; (ii) a systematic comparison of closed and open-source models in this setting; and (iii) a suite that enables the evaluation of intermediate steps, output grading, and testing robustness through controlled perturbations.

## 2. Preliminaries

To avoid ambiguity, before providing a detailed overview of our benchmark, we define the key terms related to bioinformatics and agentic evaluations, which we will use throughout the paper.

**Bioinformatics.** First, we define a bioinformatics pipeline as any workflow that transforms raw biological data (e.g., DNA/RNA sequencing reads) into analyzed outputs (e.g., alignments, expression estimates, or variant calls). Within a pipeline, we distinguish two kinds of data: input data and reference data. Input data is the sample material being processed (for example, reads from a particular experiment or individual). Reference data is the shared baseline used across many samples for steps like alignment or annotation (for example, it can be a reference genome, gene annotation, or a known variant database).

**AI agent evaluation.** Separately, when discussing our evaluations, we use a different set of terms and draw their definitions primarily from Grace et al. (2026). A task is a single instance with defined inputs, and a clear success criterion. A single execution of that task is a trial. We score a trial using a grader that compares an agent's outputs with the ground truth. During a trial, we record a transcript (also called a trace or trajectory) comprising the full log of all the actions and intermediate steps taken by an agent (e.g., tool calls, reasoning traces), and the events which occurred as a result. The outcome the final result produced by executing the entire pipeline.

Next, we define the systems which support the core infrastructure of the framework: an evaluation harness, agent harness, and an evaluation suite. An evaluation harness is the system that executes tasks, captures transcripts, applies graders, and aggregates results across trials. An agent harness (or scaffold) is the system that enables a model to take actions and use tools; in this framing, an agent is simply the model plus that harness.

Finally, an evaluation suite is a collection of tasks designed to measure specific capabilities or behaviors of agents. Tasks in a suite typically share a broad goal. In our framework,

this goal is successful and robust automation of pipelines in bioinformatics. We thus build BioAgent Bench around concrete end-to-end bioinformatics tasks and evaluate agents across multiple trials and controlled perturbations described in the following section.

## 3. Benchmark Design

BioAgent Bench (Figure 1) comprises manually curated bioinformatics tasks that span modalities and workflows common in the field: bulk and single-cell RNA-sequencing, comparative genomics, variant calling, metagenomics, viral metagenomics, transcript quantification, and experimental evolution. We frame each task as an end-to-end pipeline that requires tool orchestration, file handling, and structured reporting. This framing therefore contrasts existing frameworks that rely on single-step question answering as an evaluation unit and facilitates understanding of an agent's behavior and its failure modes. To enable automated evaluation of the results, we specify concrete ground truth outputs, typically in CSV format.

A common theme in both the paper and BioAgent Bench is verifiability. In biology, conclusions often depend on choices across the analysis pipeline, which which decreases the veracity of the conlusions drawn from the outputs. The field typically follows a canonical workflow from sample preparation and measurement, through preprocessing, to downstream analysis and statistical inference, which BioAgent Bench mirrors. In domains with small effect sizes and high sensitivity to modeling decisions, reasonable choices about design, normalization, batch correction, and statistical assumptions can yield different and even conflicting conclusions from the same data, making single-experiment outputs underdetermined. Examples include short-timescale evolution or selection experiments, subtle differential expression studies, cross-batch single-cell RNA-seq comparisons, and microbiome association analyses.

Taking these ideas into account, the intended purpose of the dataset selection process and curation is to make BioAgent Bench closer to software engineer benchmarks rather than biology data analysis benchmarks. We make this design decision primarily to enable use cases revolving around model post-training such reinforcement learning, distillation, or tweaks to harnesses. Thus, during the curation phase, we targeted tasks that are most widely used in bioinformatics, and lend themselves easily to being described in code and tool-calling pipelines.

Additionally, we impose two constraints on the potential datasets, which drastically reduce the eligible set: (1) runtime must be below 4 hours, and (2) workflows must be runnable with 48GB of RAM. By enforcing these constraints, we focus on datasets from smaller organisms where

the required reference data can be directly provided as inputs. However, these design decisions reduce fidelity to real-world practice by excluding large-organism workflows (e.g., human sequencing) and by omitting common research tasks such as finding, downloading, and staging large reference resources. Nonetheless, restricting resources requirements makes large-scale agent benchmarking feasible in the first place. Relaxing these restrictions would substantially increase infrastructure demands and evaluation complexity thereby limiting both depth and bredth of our experiments.

In Table 1, we list our ten curated tasks and their key properties. Reference implementations are written in Python (three tasks), R (two tasks), and bash (five tasks). Seven tasks require interaction with tools, while four tasks are verifiable, i.e., a pass/fail label can be assigned to their outputs. A singular task consists of: (1) a natural-language prompt that specifies the goal and expected output format (e.g. *Perform metagenomic analysis of control and fertilized samples*), and (2) the associated files required to execute and evaluate the task. The files include the primary input data and, when available, reference data (e.g., taxonomic database). This definition ties the instruction and the accompanying data together as a single unit for agent evaluation.

## 4. Experimental Setup

To evaluate each agent, we run each model in a harness. We sandbox the evaluation and tie it to a hashed run directory. As inputs to the agent, we provide: (1) a system prompt; (2) input files; (3) prompt instructions with a goal and expected output format of the outcome (see Appendix A.1–A.3). The system prompt instructs the model to generate artifacts in each step (e.g., quality control, read trimming, assembly). The generated output files and the outcome file are passed for evaluation to the grader. In our experiments we rely to GPT-5.1 as a grader which evaluates the outputs against the provided ground truth.

In our experiments, we do not compare individual LLMs directly as the downstream performance of a respective agent is heavily influenced by the harness in which it runs. Therefore, our results should be interpreted as an evaluation of agentic capabilities, i.e., the combined performance of a model and its harness.

### 4.1. Model Settings

We evaluate top-performing open- and closed-weight models from SWE-bench (Jimenez et al., 2024) at the time of running the evaluation. We run the models in three available harnesses: Claude Code (Anthropic), Codex CLI (OpenAI), and OpenCode (AnomalyInnovations). Each run uses a simple system prompt that points the LLM at the available environment, strategy for producing artifacts, and the final

*Table 1.* An overview of BioAgent Bench tasks and their key properties: language of the reference implementation (**Language**), whether the task requires interactions with tools (**Tool calls**), and whether their outputs can be scored as a binary outcome (**Verifiable**). ✓ indicates that a task in the corresponding row satisfies the property in the corresponding column, whereas ✗ indicates the opposite.

| Identifier | Name | Language | Tool calls | Verifiable |
|---|---|---|---|---|
| alzheimer-mouse | Alzheimer Mouse Models: Comparative Pathway Analysis | Python | ✗ | ✗ |
| comparative-genomics | Comparative Genomics: Co-evolving Gene Clusters | R | ✗ | ✗ |
| cystic-fibrosis | Cystic Fibrosis Mendelian Variant Identification | bash | ✓ | ✓ |
| deseq | RNA-Seq Differential Expression (DESeq2) | Python | ✓ | ✗ |
| evolution | Experimental Evolution Variant Calling (E. coli) | bash | ✓ | ✗ |
| giab | GIAB Variant Calling (NA12878) | bash | ✓ | ✓ |
| metagenomics | Metagenomics: Community Comparison (Cuatro Cienegas) | R | ✓ | ✗ |
| single-cell | Single-cell RNA-seq: Skeletal Muscle Exercise Response | Python | ✗ | ✗ |
| transcript-quant | Transcript Quantification (Simulated RNA-Seq) | bash | ✓ | ✓ |
| viral-metagenomics | Viral Metagenomics: Species Identification (Dolphin) | bash | ✓ | ✓ |

completion condition. Although we run each agent in a sandboxed folder, agents have access to the internet. We run every run with "high" reasoning effort whenever available.

### 4.2. Grader Logic

An LLM-based grader scores each trial. We opt for LLM grading for two main reasons. The first reason is that bioinformatics tasks often admit multiple valid solution paths and tool choices. For example, when conducting a germline variant calling analysis, starting from the same data, an agent can opt to use a GATK4 Haplotype caller pipeline (Poplin et al., 2018b) (a canonical pipeline of multiple steps), DeepVariant (Poplin et al., 2018a) (a deep learning model), or one of the many other germline variant calling pipelines. Each of these pipelines result in a different number of steps needed to complete the analysis. An LLM grader lets us score the outputs and steps against a rubric without hard-coding a single canonical truth, which permits variable step counts for the same task. The second reason is that bioinformatics analyses produce a large volume of intermediate files. Given the number of trials we evaluated and the nondeterminism of solution paths, a comprehensive manual review would require substantial effort from a domain expert familiar with the relevant analysis and workflow, which is impractical given the pace at which agentic tools evolve.

The grader takes as input:

- **input** and **reference** data paths
- the **expected outcome** (CSV/TSV table as text)
- the **agent's outcome** (CSV/TSV table as text if exists)
- **agent's trace** (only folders and file paths)
- **grading logic prompt** (see Appendix A.4) where the grading logic prioritizes evidence of pipeline completion over numerical accuracy

The grader outputs:

- **steps_completed**: number of pipeline steps the agent

demonstrably completed.
- **steps_to_completion**: estimated total steps required to finish the task.
- **final_result_reached**: a binary variable indicating whether the agent produced the final requested result artifact.
- **results_match**: task-specific correctness flag from rubric rules
- **f1_score**: F1-score where applicable (only giab in the current version of the framework)

### 4.3. Evaluation Suite

In addition to assessing the performance in a single trial, we measure robustness across multiple trials and evaluate the effect of controlled perturbations. More concretely, our evaluation suite supports the following settings:

- **Multiple trials** to assess the variation in outputs and trajectories
- **Prompt bloat**. Trials where the task prompt is unnecessarily inflated. We augment the task prompt with additional, topically related but non-essential text to assess robustness to distraction and instruction overload.
- Trials with **corrupted data** We synthetically corrupt selected input files and measure whether the agent detects the corruption and avoids proceeding with invalid data.
- Trials with **decoy data** which an agent should ignore. We augment the task prompt with additional, topically related but non-essential text to assess robustness to distraction and instruction overload.

Our primary metric is completion rate (%). For each task, we evaluate whether the agent completes each required pipeline step and produces the requested final artifact in the specified format (CSV or TSV). The completion rate is the percentage of required steps that pass this check, as assessed by the LLM grader.

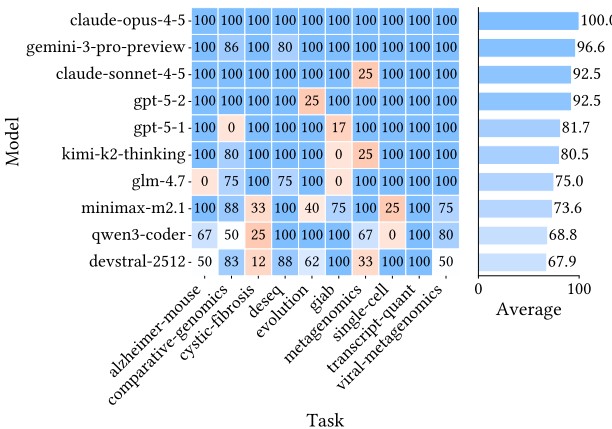

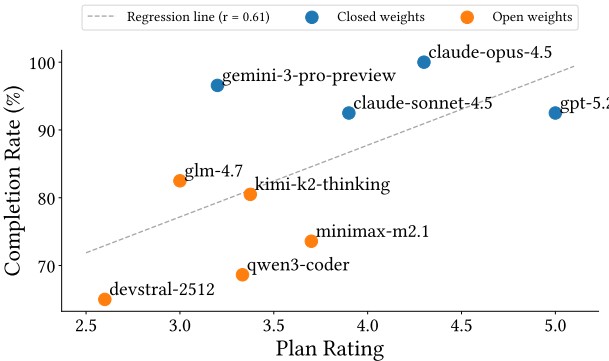

*Figure 3.* A relationship between each model's average plan quality score and its overall pipeline completion rate.

*Figure 2.* Model-task completion heatmap. The left panel shows a pairwise completion matrix: rows and columns correspond to models and tasks, respectively, and each cell reports the completion rate (in %) for each model and task pair. Cell color encodes the completion rate, with numeric annotations shown for readability. The right panel summarizes performance across tasks by reporting each model's average completion rate, providing an overall ranking of models.

## 5. Results

We present single-run pipeline completion results in Figure 2. Across al the tasks, frontier models achieve high pipeline completion rates. Claude Opus 4.5 attains a 100% completion rate, while Gemini 3 Pro, GPT-5.2, and Sonnet 4.5 each exceed 90%. The top results are obtained using the Codex CLI harness (see Appendix 4 for results across harnesses). Open-weight models trail on average, with the best-performing model, GLM-4.7, reaching 82.5% in Codex CLI, and other open-weight models ranging down to 65%.

These results suggest that current frontier models can reliably execute multi-step bioinformatics workflows end-to-end, reaching the requested final artifact without additional scaffolding. In contrast, the open-weight models evaluated here show materially lower completion rates.

To investigate whether this discrepancy is the result of a model's internal knowledge or multi-turn agentic capabilities, we instruct the models to produce a higher-level plan without execution of the pipelines with the same input data and task prompt as in the clean run. We then instruct GPT-5.1 to score each plan on a 1–5 scale. As shown in Figure 3, planning quality correlates with overall agentic performance (Pearson $R = 0.61$), suggesting that models with better plans generally translate into higher end-to-end pipeline completion success. However, this relationship is not deterministic; open-weight models tended to receive lower planning scores, yet some models (e.g., Gemini-Pro-3) produced weaker plans than frontier models while still completing pipelines at high success rates. Thus, successful completion can be achieved despite lower-quality explicit planning, in-

dicating that success depends on meeting a certain level of domain knowledge, while shortcomings in agentic capabilities remain the main bottleneck for open-weight models.

We also observed a qualitative difference in failure modes during manual inspection of a subset of runs: some closed-source models appeared more prone to getting stuck in repeated error-correction loops or terminating prematurely before reaching a solution, whereas frontier models more often recovered and completed the pipeline. We leave a systematic analysis of these behaviors to future work.

### 5.1. Robustness

To further systematically evaluate the agents' understanding of the data and the underlying tasks, as a proxy for testing the biological reasoning, we evaluate the stability of results, i.e., robustness across multiple trials. We focus our analysis on evaluated GPT-5.2 in Codex CLI harness [1]. For each task, we run the agent for four trials and compare the stability of the results using the Jaccard Index for categorical data (e.g., KEGG Pathways, Gene IDs), and the Pearson correlation coefficient for numerical data (e.g., p-values, abundance) [2]. We compute the Jaccard index of the final results across trials of the same task. When computing the Pearson correlation coefficient across trials within each task, we consider only IDs shared by all runs and drop missing rows. We finally average the correlations across all trial pairs. If no valid correlations exist, the result is NaN.

The mean Jaccard overlap is 0.43, while the Pearson correlation coefficient is 0.73, indicating that there is considerable variability in the final results across trials. We hypothesize this variability is driven by a combination of non-determinism in tool execution and between-trial differences

---

[1]We do not run robustness experiments on open-weight models because their pass@1 was low for several tasks, yielding too few fully completed trials per task to compute stability metrics reliably.

[2]For the data used in calculation, see Appendix A.6

in inferred parameters or intermediate decisions made during the pipeline (e.g., statistical inference choices). See Appendix A.7 for an isolated case study.

*Table 2.* Jaccard index and Pearson correlation across four trials per task.

| Task | Jaccard | Pearson |
|---|---|---|
| alzheimer | 0.160 | 0.219 |
| comparative | 0.004 | NA |
| cystic-fibrosis | 1.000 | NA |
| deseq | 0.978 | 0.995 |
| evolution | 0.000 | NA |
| metagenomics | 0.395 | 0.746 |
| single-cell | 0.114 | 0.395 |
| transcript-quant | 1.000 | 1.000 |
| viral-metagenomics | 0.667 | 1.000 |

## 5.2. Perturbation Analysis

As a complementary component of the evaluation suite aimed at probing biological reasoning below the level of global pipeline construction, we evaluate agent behavior under task-specific perturbations described in subsection 4.3. To characterize the agent's response to these perturbations, we manually inspect execution traces. To assess agent's behavior under the decoy condition, we retrieve all code blocks and tool invocations that reference the decoy filenames, and verify whether the decoy data were incorporated into analysis (e.g., used in commands, loaded into scripts, or included in downstream result aggregation). For the corrupted-input condition, we inspected agent messages and tool calls that referenced the corrupted filenames to determine whether the agent explicitly flagged the issue and whether it attempted to continue the pipeline regardless.

As summarized in Table 3, the agent correctly identified corrupted inputs in 7/10 tasks, decoy files were used erroneously in 2/10 tasks, while prompt bloat had a pronounced negative effect on overall completion across tasks and trials; agents completed 28% fewer steps relative to the unperturbed setting. For task-specific details about the perturbations see Appendix A.8.

To gain deeper insight into the failure modes under perturbations, we qualitatively analyze each (perturbation, task) pair. We summarize our analysis in the following paragraphs

**Corrupted data.** Upon manual inspection of the runs under the corrupted-input condition we observe several distinct failure modes. In some tasks, the agent failed to detect corruption and proceeded as if the inputs were valid. For example, in *alzheimer-mouse*, the agent continued with differential-expression analysis despite an obviously corrupted DESeq2-style distribution. In *comparative-genomics*, it again operated indiscriminately over available sequences rather than validating input integrity.

*Table 3.* Agent perturbation outcomes across tasks. Shows identified corrupt data (✓ indicates desired behavior), whether a decoy was used (✗ indicates desired behavior), and the resulting change in completion performance due to prompt bloat (percentage points; higher is better).

| Identifier | Corrupted | Decoy | Δ Completion (%) |
|---|---|---|---|
| alzheimer-mouse | ✗ | ✗ | -12.5 |
| comparative-genomics | ✗ | ✓ | -20.0 |
| cystic-fibrosis | ✗ | ✗ | 0.0 |
| deseq | ✓ | ✗ | -100.0 |
| evolution | ✓ | ✗ | 75.0 |
| giab | ✓ | ✗ | -20.0 |
| metagenomics | ✓ | ✓ | -100.0 |
| single-cell | ✓ | ✗ | -100.0 |
| transcript-quant | ✓ | ✗ | 0.0 |
| viral-metagenomics | ✓ | ✗ | 0.0 |

In other tasks, the agent did flag corruption but continued the pipeline regardless, typically by attempting to "route around" the issue via alternative analyses or reference downloads. We observed this pattern in *evolution*, *giab*, and *metagenomics*. A related phenomenon occurred in *transcript-quant*: the agent identified the corrupted input but still attempted subsequent steps until a downstream tool invocation failed due to missing or unusable outputs.

Finally, we observe cases where corruption led to early termination, either appropriately or due to cascading failures. In *deseq*, the input FASTA sequences were unusable, and the agent exited early. In contrast, in *single-cell* the agent exhibited the desired behavior by identifying the corrupted data and not proceeding with further analysis. Finally, the *cystic-fibrosis* task surfaced the following failure mode: although the metadata were scrambled and inconsistent with the prompt specification, the agent continued the analysis, and produced incorrect results. We note that these behaviors are also sensitive to the type and severity of the corruption introduced in our ablation. Some perturbations rendered files effectively unreadable, in which case downstream tool failures can force early termination irrespective of the agent's intent. Other perturbations preserved superficially valid structure (e.g., files that load or parse but contain implausible values or scrambled semantics), allowing pipelines to run to completion despite being scientifically invalid. While the latter category may not trigger hard errors, it would typically be readily apparent to a human practitioner that the inputs are not trustworthy and should not be used for inference.

**Decoy files.** A closer inspection of the decoy failures suggests two recurring patterns. First, in the comparative genomics task, file selection was implemented via a shallow filename heuristic: the agent globbed all inputs matching the `.genomic.fna` suffix and therefore unintentionally included the decoy organism alongside the intended genome. Second, in the metagenomics task, the agent selected an in-

appropriate reference database, using a viral database in place of the bacterial database required by the task. Both cases are consistent with a failure to configure the required tool in a biological context, instead relying on surface-level cues (filenames and readily available defaults) that are insufficient under adversarial or confounded inputs. In all other cases, the agent correctly discarded the decoy files.

**Prompt bloat.** Prompt bloat induces a distinct set of behavioral failures, most clearly visible in tasks with complete degradation ($-100\%$ completion). In these cases, the agent exhibited the same qualitative symptoms we observed for weaker (open-weight) models: rather than making incremental progress through tool use and stateful execution, the agent repeatedly restated the task, cycled through superficial reformulations of the instructions, and then terminated early without producing substantive intermediate artifacts.

## 6. Discussion

BioAgent Bench aims to evaluate agentic bioinformatics behavior as it occurs in practice: selecting the right inputs and references, orchestrating multi-step tool calls, writing analysis scripts, conducting statistical inference, and delivering concrete final results. Our results suggest that today's frontier agents can often execute canonical workflows end-to-end with minimal scaffolding, but that pipeline completion can substantially overestimate reliability.

**Robustness.** The observed variability between runs (e.g., different Salmon flags, differing statistical choices) reflects two realities: (1) non-determinism in agent decisions and tooling, and (2) genuine degrees of freedom in analysis pipelines. Stability metrics (Jaccard overlap, Pearson correlation) quantify how sensitive an agent is to small internal decision differences and are an important part of the overall evaluation of agent performance and behavior. In practice, we expect useful agents to become more stable when given explicit constraints (fixed tool versions, parameter templates, prespecified reference datasets) and when equipped with internal policies such as "prefer defaults unless justified by data-driven checks". BioAgent Bench can measure how well different harnesses and prompting strategies enforce these rules.

Across tasks, high completion rates indicate that agents can (1) interpret the intended goal, (2) choose the correct tools, (3) manage environments (e.g., mamba, R packages), and (4) recover from many routine tool errors. These obsereved capabilities support the view that state-of-the-art agents can act as effective workflow assistants for common bioinformatics analyses without substantial custom engineering.

**Open-weight models and their shortcomings.** Open-weight models are beginning to close the gap, but remain behind frontier closed models in this evaluation. In our runs,

the strongest open-weight models often produced workable pipelines and sometimes achieved competitive completion, yet exhibited greater variability and more frequently failed to reach the requested final artifact. One plausible explanation is that the deficit is not purely "bioinformatics knowledge", but a combination of (1) weaker end-to-end agentic competence (tool use reliability, error recovery, and state tracking over long executions), and (2) reduced ability to form and maintain high-quality execution plans. Consistent with these hypotheses, we observe a positive correlation between plan ratings and completion, suggesting that explicit planning quality is a meaningful, though not sufficient, predictor of downstream success. Crucially, the relationship is not deterministic because some models can complete tasks despite producing weaker explicit plans, indicating that agentic capabilities can compensate.

Our robustness experiments highlight thet correct higher-level pipeline construction does not imply correct step-level reasoning. A practical implication is that benchmarks (and deployments) should treat pipeline completion as a necessary but insufficient success criterion. For sensitive use cases such as in clinical diagnostics, the relevant question is not "does it produce a result?" but "can it reliably detect when it should not proceed, and can it justify choices with evidence grounded in the data and context?".

Despite lower current completion rates, open-weight models play an important role in settings where privacy is crucial. Many realistic workflows involve sensitive patient-derived sequencing data for cancer screening, proprietary reference collections, or unpublished IP where routing data to model providers may be unacceptable. In such contexts, locally deployable agents can be preferable even when they underperform on aggregate benchmarks, because they enable stronger governance and reduce organizational risk. From this perspective, improving open-weight agent performance is not only a matter of competitiveness but an enabling step for safe and compliant deployment of agentic systems in biomedical research and clinical environments.

## 7. Limitations

Desiging principled evaluations of AI agents requires extensive resources. Thus, to ensure scalable yet informative evaluations, we made necessary trade-offs in our experimental design. We outline the key limitations of BioAgent Bench, focusing on (1) the inherent subjectivity and bias of LLM-based grading, (2) the reduced real-world fidelity caused by strict runtime and memory constraints, and (3) the limited scope and sampling of our current robustness testing.

**LLM grading can be subjective and biased.** LLM graders enable scalable evaluation when multiple solution paths

are valid, but scores can depend on rubric wording, trace verbosity, and how intermediate artifacts are presented. As a result, a grader can reward trials that look plausible even when step-level reasoning is wrong, and introduce run-to-run inconsistency (e.g., differing numbers of accepted steps), which already occurs in the current suite.

**Benchmark constraints limit real-world fidelity.** Resource caps (runtime $< 4$ hours, $\leq 48$GB RAM) improve reproducibility but narrow the task and dataset space. They tend to favor smaller organisms and well-packaged inputs, underrepresenting common failure modes such as large genomes, heterogeneous cohorts, messy metadata, and long-running pipelines. The benchmark also excludes a major part of applied bioinformatics: discovering, curating, and justifying external references and best practices from primary sources.

**Robustness testing is limited in scope and sampling.** Perturbation analysis uses a single trial per task and condition, limiting uncertainty estimates and variance comparisons across models and harnesses; results should be treated as suggestive. Perturbation coverage is also narrow relative to practice, where confounders include multiple plausible files, misleading names or metadata, partial truncations, and subtle format violations. Corruption difficulty depends on construction: some cases are obvious (malformed files), while more realistic and safety-relevant cases remain syntactically valid but biologically implausible.

## 8. Related Work

**General LLM agent benchmarks.** The evaluation of large language models as autonomous agents has emerged as a critical research direction, moving beyond traditional static benchmarks to assess models in dynamic, interactive environments. AgentBench (Liu et al., 2024) provides a comprehensive evaluation framework spanning eight distinct environments, including operating systems, databases, knowledge graphs, and web browsing. Aiming to evaluate the abilities of LLMs to effectively utilize external tools and APIs Li et al. 2023 design a benchmark comprising 73 APIs and 314 tool-use dialogues, and decompose evaluation into three progressive levels: API call accuracy, retrieval capability, and end-to-end task completion with planning. ToolBench (Qin et al., 2024b;a; Guo et al., 2024) further broadens this scope by covering 16k real-world APIs across 49 categories. Our work is most closely related to SWE-bench (Jimenez et al., 2024; Deng et al., 2025), which focuses on software engineering and requires the models to generate patches that resolve problems reported in GitHub issues.

Compared to software engineering or general tool-use benchmarks, biomedical tasks are harder to evaluate au-

tomatically because many candidate solutions ultimately require wet-lab experiments and clinical validation. Moreover, biological systems exhibit substantial noise and heterogeneity (e.g., batch effects, protocol variation), so multiple analysis pipelines or hypotheses can be reasonable, and success is better characterized by decision quality under constraints than by a single correct output. These properties complicate benchmark construction, where the tasks must balance realism with automatic evaluation.

**Biomedical LLM benchmarks.** A growing body of work focuses on building benchmarks for LLMs and LLM-based agents for drug discovery and biomedical data analysis tasks. For example, BioML-bench (Miller et al., 2025) evaluates tasks where agents must parse a task description, build a pipeline, implement models, and submit predictions graded by established metrics across domains such as protein engineering, single-cell omics, biomedical imaging, and drug discovery. Broader capability-oriented evaluations include LAB-Bench (Laurent et al., 2024), a large multiple-choice benchmark covering practical biology research skills, including literature reasoning, database navigation, figure interpretation, and sequence manipulation. Liu et al. 2026 propose an evaluation framework to assess agent capabilities in single-cell omics analysis. Finally, BixBench (Mitchener et al., 2025) focuses on data analysis in computational biology, presenting agents with real-world scenarios that require dataset exploration, multi-step analysis, and nuanced interpretation.

In contrast to benchmarks centered on data analysis and broad research text response-based capabilities, our benchmark emphasizes multi-step bioinformatics pipelines across different bioinformatics domains and computational environments.

## 9. Conclusion

BioAgent Bench provides an end-to-end benchmark and an evaluation suite for bioinformatics agents, capturing realistic workflows that require tool orchestration, artifact production, and structured outputs. Frontier agents complete canonical pipelines with high success rates without heavy scaffolding, but robustness tests show that it comes with brittle step-level behavior such as shallow file selection heuristics, weak input validation, and sensitivity to distraction. By making these failure modes measurable, BioAgent Bench shifts evaluation from *Can it produce correct outputs?* to *Can it produce correct outputs reliably, for the right reasons, while making correct decisions throughout the pipeline?*.

To further strengthen BioAgent Bench, in future work, we aim to expand task and dataset diversity (including larger and messier inputs), add tasks that require sourcing and justifying external references, and strengthen robustness

evaluation with richer perturbations and automated scoring. More concretely, we aim to integrate robustness into the primary metrics. We expect BioAgent Bench to accelerate the development powerful locally deployable agents, capable of reducing the manual overhead in designing bioinformatics workflows under strict privacy constraints.

## Impact Statement

This paper presents BioAgent Bench, a benchmark dataset and evaluation suite for end-to-end bioinformatics agent workflows. A central intended impact is to improve agent evaluation for practical use cases. A second intended impact is to support the development of private, locally deployable agentic systems using open-weight models, by enabling standardized benchmarking to track the capabilities gap between open-source and the closed frontier. We also expect BioAgent Bench to be useful as a target for improving agentic behavior via methods such as fine-tuning, distillation, and reinforcement learning on verifiable multi-step tasks.

## Acknowledgments

This work was funded by the European Union – NextGenerationEU, project NPOO.C3.2.R2-I1.04. Marko Čuljak acknowledges support from the Croatian Science Foundation (HRZZ) Young Researchers' Career Development Project, grant DOK-NPOO-2023-10-1392 and Coefficient Giving.

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

# A. Appendix

## A.1. System prompt

```
You are an expert bioinformatics agent that assists users with bioinformatics and
computational biology tasks.  You are an expert in genomics, transcriptomics,
proteomics and related -omics domains, and you follow best practices from
the field.  Environment Management You are already working inside the mamba
environment named {env_name}.  Never attempt to activate the base environment;
keep using {env_name} unless instructed to create a new one.  Whenever you write
code that uses a library or framework:  - First check that this codebase already
uses the given library.  - Use the 'mamba list' command to check if the library
is installed.  - Install any missing libraries or packages using mamba.  - Always
install packages into the bioinformatics environment.
Code Style For making network requests or large loops add a progress meter.  Do
not keep downloaded data in memory; export it to files so you don't have to
download again.  In case of errors, wrap subprocess calls in try/except blocks
and output the exception error so you know exactly why the subprocess call failed.
Example:  result = subprocess.run(['ls', '-l'], capture_output=True, text=True)
except subprocess.CalledProcessError as e:  print(f'Error:  e.stderr') When
working with file paths, always use absolute paths rather than relative.
Task execution If inputs are missing/ambiguous, try to derive them using
other tools.  Write outputs into stage-scoped directories under special
directory 'outputs/'.  Example:  'outputs/0_trimming/' 'outputs/1_alignment/'
'outputs/3_postprocessing/' Before starting the task run the 'tree' command to
see which files have been generated.  If later you are running the tree command
you should ignore .snakemake directories.  Example:  tree -I '.snakemake' Don't
just use the integers for enumerating the steps; also describe the steps.  Example:
'0_processing'
Finalizing the pipeline Output the final results in the same format as was asked
by the user in the provided example.  Output the final results into a results/
directory.  Once the final result has been generated and placed into the results/
directory, you should stop executing the task.
```

## A.2. Task prompts

- **alzheimer-mouse**: Perform a comparative differential expression analysis of three different Alzheimer's Disease mouse models (5xFAD, 3xTG-AD, and PS3O1S) to identify shared molecular KEGG pathways. The output should be a CSV file with the following columns: `pathway`, `5xFAD_pvalue`, `3xTG_AD_pvalue`, `PS3O1S_pvalue`.
- **comparative-genomics**: Reconstruct phylogeny and identify COGs across four *Micrococcus* genomes; filter clusters present in all genomes, coding-only, with high-confidence annotations. The output should be a CSV file with the following columns: `cluster_number`, `consensus_annotation`.
- **cystic-fibrosis**: Find the genetic cause of Cystic fibrosis; identify the causal recessive variant consistent with affected siblings NA12885, NA12886, and NA12879. The output should be a CSV file with the following columns: `chromosome`, `position`, `variant_id`, `reference`, `alternate`, `gene_name`, `gene_id`, `annotation`, `impact`, `transcript_id`, `hgvs_c`, `hgvs_p`, `clinical_significance`, `diseases`, `review_status`, `rs_id`.
- **deseq**: Identify differentially expressed genes between planktonic and biofilm conditions of *Candida parapsilosis*. The output should be a CSV file with the following columns: `gene_id`, `log2FoldChange`, `pvalue`, `padj`.
- **evolution**: Identify and annotate genome variants in two evolved lines relative to an ancestor lines of *E. coli*; report only variants shared by both evolved lines of moderate or higher predicted severity. The output should be a CSV file with the following columns: `chrom`, `pos`, `ref`, `alt`, `gene`, `impact`, `effect`, `status`.
- **metagenomics**: Perform metagenomic analysis of control (JC1A) and fertilized (JP4D) samples to classify microbial taxa and estimate relative abundances. The output should be a CSV file with the following columns: `OTU`, `Kingdom`, `Phylum`, `JP4D`, `JC1A`.
- **single-cell**: Analyze single-cell RNA-seq data from pre- and post-exercise skeletal muscle samples. Perform clustering, cell type identification, and differential expression analysis within each cell type between conditions. The output should be a CSV file with the following columns: `cluster_id`, `predicted_cell_type`, `gene_name`,

`logfoldchanges, pvals, pvals_adj, direction, abs_logfc.`
- **transcript-quant**: Perform transcript quantification on the provided paired-end RNA-Seq reads using the transcriptome reference. The output should be a `.tsv` file with the following columns: `transcript_id, count`.
- **viral-metagenomics**: Analyze paired-end metagenomic sequencing data from a dolphin fecal sample to identify potential viral agents. Assemble and classify contigs, then summarize results by taxonomic domain and species. The output should be a CSV file with the following columns: `contig_count, domain, species`.

## A.3. Task description

- **alzheimer-mouse** (Peikon): Alzheimer Mouse Models: Comparative Pathway Analysis. Analyze 5xFAD, 3xTG-AD, and PS301S mouse models: normalize counts, perform differential expression, run KEGG pathway enrichment, and compare shared pathways across models.
- **comparative-genomics** (Lakshman, 2024): Comparative Genomics: Co-evolving Gene Clusters. The datasets consists FASTA sequences and GFF annotations of a microbial genome for *Micrococcus*. The goal of is to do phylogenetic reconstruction of clusters of orthologous co-evolving genes; identify functionally conserved gene clusters across the genomes and group them into co-evolving functional modules.
- **cystic-fibrosis** (Cingolani et al., 2012): Cystic Fibrosis Mendelian Variant Identification. The sample dataset is a simulated dataset for finding the genetic cause of Cystic fibrosis. The dataset is real sequencing data from CEPH_1463 dataset provided by the Complete Genomics Diversity Panel. It consists of sequencing of a family: 4 grandparents, 2 parents and 11 siblings. A known Mandelian disease mutation has been added on three siblings, taking care to be consistent with the underlying haplotype structure. The goal is to find the mutation causing the Mendelian recessive trait - Cystic Fibrosis.
- **deseq** (Holland et al., 2014): RNA-Seq Differential Expression (DESeq2). The dataset consists of RNA-Seq samples from *Candida parapsilosis* wild-type (WT) strains grown in planktonic and biofilm conditions, generated as part of a study on gene expression and biofilm formation. The samples were sequenced on the Illumina HiSeq 2000 platform. The goal of this analysis is to perform differential expression analysis using DESeq2 to identify genes that are significantly up- or down-regulated between planktonic and biofilm conditions, providing insights into biofilm-associated transcriptional changes.
- **evolution** (Schmeier, 2020): Experimental Evolution Variant Calling (E. coli). The experiment follows a similar strategy as in what is called an experimental evolution experiment. The final aim is to identify the genome variations in evolved lines of *E. coli*. The data is composed of a single ancestor line and two evolved lines. The data is from a paired-end sequencing run data from an Illumina HiSeq. This data has been post-processed in two ways already. All sequences that were identified as belonging to the PhiX genome have been removed. Illumina adapters have been removed as well already.
- **giab** (Zook et al., 2014): GIAB Variant Calling. The GIAB dataset consists of Agilent SureSelect v7 exome sequencing (∼75M reads) from the NA12878 reference sample, providing high-confidence benchmark variants on GRCh38. The task is to perform germline variant calling on NA12878.
- **metagenomics** (Zirión-Martínez et al., 2024): Metagenomics: Community Comparison (Cuatro Ciénegas). The metagenomics dataset consists of sequencing reads from the Cuatro Ciénegas Basin, comparing microbial communities under control (JC1A) and nutrient-enriched (JP4D) conditions. The task is to profile taxonomic composition and report relative abundances of bacterial taxa.
- **single-cell** (Lovrić et al., 2022): Single-cell RNA-seq: Skeletal Muscle Exercise Response. The single-cell RNA-seq dataset consists of human skeletal muscle samples collected before and after acute exercise, sequenced with 10X Genomics. The task is to identify cell types and determine their transcriptional responses to exercise.
- **transcript-quant** (Wratten et al., 2021): Transcript Quantification (Simulated RNA-Seq). The goal is to quantify transcript expression levels from paired-end RNA-Seq reads (`reads_1.fq.gz, reads_2.fq.gz`) using the provided reference transcriptome (`transcriptome.fa`). Because the data is simulated, the quantification should exactly reproduce the underlying counts. The results represent a mapping from transcript IDs → read counts.
- **viral-metagenomics** (Gourlé): Viral Metagenomics: Species Identification (Dolphin). The viral metagenomics dataset consists of paired-end sequencing reads from a dolphin with gastroenteritis of suspected viral origin. The task is to identify viral species present in the fecal sample by assembling and classifying contigs.

## A.4. Grading logic prompt

```
You are a strict, impartial Bioinformatics Pipeline Judge.  Your job is to
evaluate an LLM agent's work for executing a bioinformatics pipeline instructed
by the prompt.  The LLM agent was given an instruction to output each processing
step in a separate folder.  The data to evaluate each agent is given as follows:
 1. You are given the paths of the input and the reference data which the agent was
    given to work with.
 2. You are given the whole directory structure of the agent's work and it is your
    job to estimate how close to completing the pipeline the agent came.
 3. You are given the final results which the agent was instructed to produce, if
    they exist
 4. You are givne the truth data which is the expected output of the prompted
    pipeline.
 5. You are given the prompt which the agent was given to complete.
Inputs (provided at evaluation time)
   • 1.  Input data:  {input_data}
   • 2.  Reference data:  {reference_data}
   • 3.  Processing tree:  {processing_tree}
   • 4.  Results:  {results}
   • 5.  Truth:  {truth}
   • 6.  Prompt:  {task_prompt}
Evaluation rules:
   • Priortize evaluation of the pipeline completion over the correctness of the
     results.
   • If gene names are of different naming conventions, the result is still
     considered valid.
   • For estimating the number of steps to completion, try to estimate which
     bioinformatic-relevant steps are should be completed.
   • Count upstream steps only if their expected artifacts are present (e.g.,
     MultiQC, count matrix, indexing files).
   • Don't count placeholders or mock completion as a completed steps.
   • For example think about p-values, logfold values, or other statistics if
     present.
   • To be sure that there's no mocking or hallucinated values, make sure that prior
     steps have been generated.
   • {results_match_guidance}
Metrics to return
 1. steps_completed:  int --- The number of steps that the agent completed.
 2. steps_to_completion:  int --- The number of steps that the agent was expected
    to complete.
 3. final_result_reached:  bool --- Whether the agent reached the final result.
 4. notes:  str --- Summarize where the agent stopped if stopped and what steps are
    left to be done.
 5. results_match:  bool --- Set to true/false (or 1/0) per the rule above.
You are supposed to return the metrics as a JSON object with fields that satisifes
the schema:  EvaluationResults
```

## A.5. Harness evaluation

Table 4 shows completions rates per model and task for different harnesses.

*Table 4.* Model completion rates by harness.

| Model | codex-cli | claude-code | opencode |
|---|---|---|---|
| claude-opus-4-5 | 100.00 | 96.00 | 93.33 |
| claude-sonnet-4-5 | 92.50 | 90.00 | 83.33 |
| gpt-5-1 | 38.50 | | |
| gpt-5-1-codex | 74.67 | | |
| gpt-5-1-codex-max | 81.67 | | |
| gpt-5-2 | 92.50 | | 87.50 |
| devstral-2512 | 47.92 | | 65.00 |
| gemini-3-pro-preview | 96.57 | | 67.98 |
| glm-4-7 | 82.50 | | 75.00 |
| kimi-k2-thinking | 65.67 | | 80.50 |
| minimax-m2.1 | 12.50 | | 73.58 |
| qwen3-coder | 0.00 | | 62.92 |

## A.6. Robustness metrics

Categorical and numerical values used for calculation of Jaccard index and Pearson correlation for evaluating robustness across multiple trials of the same task.

*Table 5.* Columns used to compute Jaccard overlap (IDs) and Pearson correlation (values) for each task

| Task | Jaccard columns (IDs) | Pearson columns (values) |
|---|---|---|
| alzheimer | `pathway` | `3xTG_AD_pvalue,` `5xFAD_pvalue, PS3O1S_pvalue` |
| comparative | `consensus_annotation` | *none (Pearson not computed)* |
| cystic-fibrosis | `chromosome, position, reference, alternate` | *none (Pearson not computed)* |
| deseq | `gene_id` | `baseMean, log2FoldChange, padj, pvalue` |
| evolution | `chrom, pos, ref, alt` | *none (Pearson not computed)* |
| metagenomics | `Phylum` | `JC1A, JP4D` |
| single-cell | `predicted_cell_type, gene_name` | `abs_logfc, logfoldchanges, pvals, pvals_adj` |
| transcript-quant | `transcript_id` | `count` |
| viral-metagenomics | `domain, species` | `contig_count` |

## A.7. Robustness case study

To probe potential sources of variability, in the simplest task - **transcript-quant** we manually compared four trial logs; : *otlp-04c96955*, *otlp-14905b2e*, *otlp-bfbd6a48*, *otlp-f1d8fc0b*. Across all trials, the high-level workflow is consistent and correct: Salmon builds an index from the transcriptome and then quantifies directly from paired FASTQ inputs.

The main differences were the optional flags enabled within Salmon:

- *otlp-04c96955* and *otlp-14905b2e*: baseline runs using `--validateMappings` without bias-correction flags.
- *otlp-bfbd6a48*: enables `--keepDuplicates` during indexing; quantification remains baseline.
- *otlp-f1d8fc0b*: enables `--gcBias` and `--seqBias` during quantification (in addition to `--validateMappings`).

In summary, two runs are baseline, one differs at indexing, and one enables bias correction at quantification, which shows that there was no consistency across trials. These configuration differences provide a plausible mechanism for between-trial variability in downstream estimates.

## A.8. Perturbation settings

We evaluated robustness with three perturbation settings: prompt bloat, corrupt inputs, and decoy inputs.

**Prompt bloat.**    We generated long, task-specific but irrelevant background text. During perturbation tests, the corresponding block is prepended to the task prompt, increasing token count without adding task-critical instructions. The additional word counts per task are:

| Task | Additional words |
|------|------------------|
| alzheimer_mouse | 1271 |
| comparative_genomics | 1877 |
| cystic_fibrosis | 1452 |
| deseq | 832 |
| evolution | 1070 |
| giab | 1066 |
| metagenomics | 871 |
| single_cell | 958 |
| transcript_quant | 808 |
| viral_metagenomics | 1362 |

An example of a bloated prompt for the **metagenomics** task:

```
The metagenomics dataset consists of sequencing reads derived from environmental
samples collected in the Cuatro Ciénegas Basin, a uniquely structured aquatic and
sedimentary ecosystem known for hosting diverse microbial life across fine-scale
nutrient gradients.  In this task, you are comparing microbial communities under
two conditions:  a control condition (JC1A) and a nutrient-enriched condition
(JP4D). While both sets of reads originate from the same broader ecological
setting, the key idea is that nutrient enrichment can shift which microbes thrive,
which decline, and how the overall community composition redistributes across
taxonomic groups.
Metagenomics, in the broad conceptual sense, is the study of genetic material
recovered directly from a mixture of organisms present in a sample.  Unlike
approaches that focus on a single isolated organism, metagenomics aims to
capture a ``community snapshot'' of many microbes at once---often bacteria, but
potentially also archaea, viruses, and microbial eukaryotes depending on the
sample and what is detectable.  In practice, the raw material you start with is
a large collection of short fragments of biological sequence information (reads)
that collectively reflect the organisms whose DNA (or genetic material) was
present in the sampled environment.  Because environmental samples contain many
organisms simultaneously, the reads are interleaved:  fragments from different
taxa are mixed together, and the analysis goal is to infer ``who is there'' and
``in what proportions,'' rather than reconstructing one single genome.
A central concept in community metagenomics is taxonomic composition, which
refers to the list of taxa detected and their relative representation.  A taxon
is a named biological grouping such as a species, genus, family, order, class,
or phylum.  Depending on the data and how confidently reads can be attributed,
taxonomic assignments may be made at different ranks.  It is common in microbial
community summaries to report abundances at a rank that is both informative and
reasonably stable---often genus or family---though the appropriate rank can vary
based on the confidence and granularity of classification.  Importantly, taxa are
not merely labels:  they are a structured hierarchy, meaning that two distinct
species may belong to the same genus, and multiple genera may belong to the same
family, and so on.  When comparing communities, shifts may be visible at multiple
ranks, and sometimes signals that are subtle at species-level become clearer at
higher ranks.
Another core concept is relative abundance.  In most metagenomics community
profiling settings, especially when comparing conditions, we care about how
large a fraction of the community is represented by each taxon relative to the
```

```
total.  Relative abundance is typically expressed as a proportion or percentage:
for example, ``Taxon X constitutes 12% of detected bacterial reads in JC1A.''
Relative abundance is valuable because it supports comparisons even when the total
amount of sequence data differs between samples, as it normalizes by the total
measured signal.  However, relative abundance is also inherently compositional:
if one taxon's relative abundance increases, others must collectively decrease
to keep the total at 100%.  This means interpretations should focus on changes
in community composition rather than assuming that every increase corresponds to
absolute growth (unless absolute measurements are provided, which they are not
here).
When thinking about nutrient enrichment in microbial ecosystems, it is helpful
to frame it as a broad ecological perturbation.  Nutrients can act as limiting
resources; when they become more available, microbes that can quickly exploit them
may become more prevalent, potentially outcompeting taxa adapted to nutrient-poor
conditions.  Conversely, taxa that are specialized for low-nutrient environments
may decline in relative representation if the enriched environment favors
different metabolic strategies.  Nutrient changes can also indirectly influence
community structure by shifting interactions such as cross-feeding, competition,
and niche partitioning.  Importantly, the task does not require any narrative
ecological interpretation; it focuses on accurately describing which bacterial
taxa are present and their relative abundances in each condition.
It is also useful to understand what is meant by ``profiling'' in this context.
Profiling means producing a structured inventory of bacterial taxa detected in
each sample condition and quantifying their relative abundances.  It does not
imply mechanistic interpretation, functional annotation, or inference of metabolic
pathways.  The objective is not to speculate about why a taxon is present, nor to
infer environmental parameters.  Instead, it is an exercise in summarizing and
comparing community membership and proportional representation in a consistent
way.
The term ``bacterial taxa'' indicates that the report should focus specifically on
bacteria rather than other biological groups.  In mixed microbial datasets, there
can be signals from non-bacterial sources; however, the reporting target here is
bacterial composition.  Conceptually, bacteria are one of the dominant microbial
domains in many environmental communities, and bacterial taxonomic summaries are a
standard output for metagenomics comparisons.  The report should reflect bacterial
taxa and their relative abundances, capturing the compositional profile of the
community for each condition.
Because the data come from natural samples, it is worth keeping in mind some
general properties of environmental sequencing-derived read collections.
Environmental samples can contain uneven distributions of organisms:  a small
number of taxa may dominate while many others are present at low levels.
Additionally, detection and apparent abundance can be influenced by biological
and sampling factors such as community heterogeneity and stochastic sampling of
rare taxa.  Again, for this task, these are background considerations to support
careful reporting, not instructions to perform any specific procedure.
```

**Corrupt inputs.** We synthetically corrupt selected input files and measure whether the agent detects the corruption and avoids proceeding with invalid data.

- **alzheimer-mouse**: synthetic differential expression table with constant/uninformative values (DEA_PS3O1S.csv;
- **comparative-genomics**: inserted 1600 A bases in random places in 4 `.fna` files.
- **cystic-fibrosis**: scramble the metadata.
- **deseq**: `.fastq` files with $\sim 90\%$ bases replaced by N and all quality scores set to `"!"` (Phred 0).
- **evolution**: ancestor `*.fastq.gz` (only files containing `"anc"`) with $\sim 90\%$ bases replaced by N and all quality scores set to `"!"` (Phred 0).
- **giab**: `*.fastq.gz` with $\sim 90\%$ bases replaced by N and all quality scores set to `"!"` (Phred 0).
- **metagenomics**: `*.fastq.gz` with $\sim 90\%$ bases replaced by N and all quality scores set to `"!"` (Phred 0).
- **single-cell**: `.mtx` entries rewritten to constant 777.
- **transcript-quant**: `*.fastq.gz` with $\sim 90\%$ bases replaced by N and all quality scores set to `"!"` (Phred 0).
- **viral-metagenomics** — `*.fastq.gz` with $\sim 90\%$ bases replaced by N and all quality scores set to `"!"` (Phred 0).

**Decoy inputs.** We inject decoy inputs (e.g., sequences from an unrelated organism) to test whether the agent can correctly contextualize the data and exclude irrelevant files from downstream analysis.

- **alzheimer-mouse**: inserted synthetic mouse differential expression table with randomized gene ids/names and randomized statistics.
- **comparative-genomics**: inserted genomic sequence from E.Coli
- **cystic-fibrosis**: inserted random VCF
- **deseq**: inserted genomic sequence and scaffold from Candida tropicalis strain
- **evolution**: inserted control library .fastq.gz with N-only reads (length 150) and Phred 0;
- **giab**: inserted NA12877_R1/R2.fq.gz 150k read pairs (150 bp) from GRCh38 reference
- **metagenomics**: inserted a Kraken viral database.
- **single-cell**: inserted cell marker metadata with obviously wrong values.
- **transcript-quant**: inserted short random transcriptome FASTA.
- **viral-metagenomics**: inserted reference genome for Delphinapterus Leucas

