# OpenReview forum: "BioAgent Bench: An AI Agent Evaluation Suite for Bioinformatics"
_ICML.cc/2026/Conference — ICML 2026 regular_

### Official Review · Reviewer_851Q · 2026-03-02

**Soundness:** 2
**Presentation:** 2
**Significance:** 3
**Originality:** 3
**Overall Recommendation:** 4
**Confidence:** 3

**Summary:**

This paper introduces BioAgent Bench, a benchmark dataset and evaluation suite designed to assess the performance and robustness of LLM agents on end-to-end bioinformatics workflows. The benchmark comprises 10 curated tasks spanning domains like RNA-seq, variant calling, and metagenomics. Instead of multiple-choice questions, tasks require agents to orchestrate tools and produce concrete output artifacts, such as CSV or TSV files. The authors evaluate 10 models (open and closed-weight) across three agent harnesses using an LLM judge (GPT-5.1) to score pipeline progress. While frontier models achieve high completion rates, robustness tests utilizing corrupted inputs, decoy files, and prompt bloat reveal brittle step-level reasoning. The benchmark highlights the performance gap between open and closed models, emphasizing the necessity of capable local models for privacy-constrained clinical settings.

**Compliance With Llm Reviewing Policy:**

Affirmed.

**Final Justification:**

Thank you for the detailed response and for taking the time to address my concerns.

While I understand your points—especially regarding the practical challenges of visualizing large agent trajectories—I still feel that the paper's analysis and presentation are a bit too shallow. In my view, simply collecting data and running models across a pipeline does not provide quite enough of a deep-dive contribution to the field.

This is particularly true for the biology aspect of the paper. Since the tasks involve well-known concepts like DESeq, evolution, and others, there is a great opportunity to extract deeper, domain-specific insights. It would be highly valuable to analyze and visualize exactly why models fail, what specific biases they show on real-world tasks, or what common vulnerabilities are exposed by this benchmark.

Overall, while the paper serves as a functional benchmark, I still believe it lacks the necessary experimental visualizations to truly ground the findings (particularly from the right half of page 7 through page 8).

For these reasons, I will be maintaining my original score of Weak Accept. Thank you again for your hard work, and best of luck with the paper.

**Key Questions For Authors:**

- LLM Judge Criteria: Why did the evaluation design explicitly deprioritize the correctness of the final results in favor of pipeline completion (as seen in the Appendix A.4 prompt)? Given that the expected outcomes are highly structured CSV/TSV files, why not employ a strict, deterministic script (e.g., exact match, numerical tolerance checking, or standard F1-scores for all tasks) to evaluate the final artifacts independently of the LLM judge's subjective interpretation?


- Case Studies and Visualizations: The Discussion (Section 7) is overly abstract. Can you provide qualitative visual analyses or deep-dive case studies of agent trajectories in the main text? For example, a visual side-by-side of a successful versus failed tool-use trace would greatly ground the numerical results.

- Harness-Level Analysis: As an evaluation suite, the analysis of the agent harnesses is relatively shallow. Can you provide an analysis of dimensions at the harness/CIO level, such as specific tool-use distributions, skill invocation, MCP (Model Context Protocol) effectiveness, or the potential for multi-agent (Agent Swarm) collaboration?

- Perturbation Variance: Given the inherent non-determinism of agent trajectories discussed in Section 6.1 , how can we be statistically confident that the failures observed in Table 3 are due to the perturbations rather than standard run-to-run variance, considering only one trial was run per condition?

**Limitations:**

yes

**Strengths And Weaknesses:**

## Soundness:

### Strengths:

Framing the benchmark around end-to-end pipeline execution rather than static QA is highly realistic for computational biology. The inclusion of perturbation tests (decoys, data corruption) is a rigorous way to separate syntax-matching from actual domain reasoning.

### Weaknesses:

1.  Over-reliance on the LLM judge: The evaluation explicitly instructs the GPT-5.1 judge to "Priortize evaluation of the pipeline completion over the correctness of the results". In bioinformatics, producing a functionally running pipeline that outputs scientifically incorrect numbers is a failure. By downgrading exactness in favor of "plausible completion," the benchmark risks inflating the performance of models that are good at writing boilerplate code but poor at producing valid science.

2.  Rough experimental validation: The perturbation analysis relies on a single trial per task and condition. Given the high run-to-run variability demonstrated in the authors' own stability analysis, a single trial is statistically insufficient to separate a perturbation-induced failure from standard variance.

3.  Shallow harness analysis: Although framed as an evaluation harness/suite, the paper treats the harnesses largely as black boxes. It lacks an in-depth analysis of tool usage efficiency, skill retrieval, Model Context Protocol (MCP) integrations, or multi-agent architectures (Agent Swarms).

## Presentation:

- Strengths: Figure 1 provides an excellent visual overview of the evaluation harness and its inputs/outputs. The problem formulation is clear.

- Weaknesses: The later sections of the paper—specifically the Discussion (Section 7) and Conclusion (Section 9)—lack substantive analytical depth and read somewhat like filler. The paper is heavily reliant on aggregate numerical metrics and desperately needs qualitative visualizations of the agent trajectories and detailed case studies in the main text. Relegating a brief textual case study to Appendix A.7  is insufficient; the main text should visually break down where and why an agent failed or succeeded at a granular level.

## Significance:
Strengths: Bioinformatics is a prime target for agentic automation due to the tedious nature of chaining command-line tools. The authors make a highly practical and significant point regarding the necessity of open-weight models for handling sensitive patient sequencing data.
## Originality:

Strengths: Adapting agent tool-use benchmarks to verifiable bioinformatics artifacts is a valuable contribution. The explicit testing of prompt bloat and decoy files to evaluate biological context-grounding is a fresh perspective for this domain.

---

> ### Author Rebuttal · Authors · 2026-03-28
>
> We thank the reviewer for the detailed and constructive review. We especially appreciate the recognition that end-to-end pipeline execution is the right level of abstraction for evaluating realistic computational biology agents, and that the perturbation tests help distinguish superficial tool use from more grounded reasoning.
>
> **[W1, Q1] Grading design and deterministic evaluation**
> We agree that scientifically incorrect outputs should count as failures in bioinformatics, and our intent is not to treat “plausible completion” as sufficient scientific success. The reason the grading prompt prioritizes evidence of pipeline completion is that many benchmark tasks admit multiple valid workflows, making a single deterministic evaluator brittle. Several tasks are inherently underdetermined: different reasonable pipelines can produce different but still scientifically defensible outputs. In such cases, an exact string match or a single hardcoded numerical target can incorrectly penalize valid solutions. Importantly, the grader does not judge result correctness arbitrarily. Expert specified research goals, output formats and the intended result characteristics are already encoded in the grader design, so the grader evaluates whether the agent’s outputs and execution trace satisfy these expert defined requirements. It is also worth noting that strict graders often became too strict, penalizing valid runs because of formatting differences, nomenclature differences, or small methodological variations that do not meaningfully invalidate the biological result. To give one concrete example: two correct pipelines may use different reference databases or conventions, leading to output labels such as “Unclassified” versus “Unassigned". A deterministic grader can incorrectly mark one of these as a failure despite the underlying biological conclusion being correct. Our design choice therefore reflects the tradeoff of a more flexible grader paired with targeted perturbation tests. We agree this is a tradeoff and not a perfect solution, but we believe it is the more appropriate choice for the current benchmark design.
>
> **[W2, Q4] Perturbation analysis and robustness claims**
> We agree that a single trial per perturbation condition is not sufficient for a statistically comprehensive robustness estimate. This is especially true for the prompt bloat setting where we see some performance degradation which could be explained with statistical variability, but in cases where there is full degradation, meaning no valid artifacts/steps have been produced, we can infer that the prompt bloat was the culprit. Meanwhile in the perturbation tests with decoy and corrupt files, we are directly looking into the logs if those files have been used for the analysis, which is informative without elaborate statistics. A practical challenge that we experienced here is that perturbation analysis is difficult to fully automate. Agent trajectories generate many files, context sizes become large, logs are lengthy, and identifying the exact reason for failure requires careful inspection of traces. We therefore had to rely in part on manual review. Moreover, even seemingly straightforward checks can be misleading, a decoy file may not appear explicitly in a tool call log, but the agent may still have processed it indirectly inside a loop over matching file extensions. This is part of why repeated perturbation trials were beyond the practical scope of the current paper, even though we agree they would strengthen the statistical claims.
>
> **[W3, Q3] Scope of contribution - Harness evaluation**
> Our contribution is the design and analysis of the evaluation suite, of which the agent harness is one component, rather than a paper primarily about harness architecture itself. Accordingly, our goal was not to benchmark or dissect harness design choices such as tool use efficiency, skill retrieval, MCP integrations, or multi-agent/swarm coordination as standalone research questions. The released evaluation suite is designed to be modular, so that graders, harnesses, models, MCP integrations, and multi-agent variants can be swapped in and studied in future work.
>
> **[Q2] Qualitative visualizations and case studies**
> Qualitative visualizations and trajectory level case studies would make the discussion more concrete and accessible. In the current version, we included only a brief case study from a robustness testing trial, but we agree that more detailed analyses of successful versus failed traces would strengthen the paper and better ground the aggregate results. As explained above, a large practical challenge is that the trajectories and logs are often very large which makes it very time consuming to include representative visualizations in a concise way at the right level of abstraction without detailed manual review, while automatic review would overlook many of the possible edge cases.

---

> > ### Author Rebuttal · Reviewer_851Q · 2026-04-02
> >
> > Thank you for the detailed response and for taking the time to address my concerns.
> >
> > While I understand your points—especially regarding the practical challenges of visualizing large agent trajectories—I still feel that the paper's analysis and presentation are a bit too shallow. In my view, simply collecting data and running models across a pipeline does not provide quite enough of a deep-dive contribution to the field.
> >
> > This is particularly true for the biology aspect of the paper. Since the tasks involve well-known concepts like DESeq, evolution, and others, there is a great opportunity to extract deeper, domain-specific insights. It would be highly valuable to analyze and visualize exactly *why* models fail, what specific biases they show on real-world tasks, or what common vulnerabilities are exposed by this benchmark.
> >
> > Overall, while the paper serves as a functional benchmark, I still believe it lacks the necessary experimental visualizations to truly ground the findings (particularly from the right half of page 7 through page 8).
> >
> > For these reasons, I will be maintaining my original score of Weak Accept. Thank you again for your hard work, and best of luck with the paper.

---

> > > ### Author Response · Authors · 2026-04-07
> > >
> > > Thank you for your thoughtful review and for highlighting the importance of deeper biology-specific analysis and stronger visualizations to better ground the findings. We appreciate these suggestions and will keep them in mind as we continue developing this line of work.

---

### Official Review · Reviewer_mcnT · 2026-03-10

**Soundness:** 3
**Presentation:** 3
**Significance:** 2
**Originality:** 2
**Overall Recommendation:** 3
**Confidence:** 3

**Summary:**

BioAgent Bench is a benchmark dataset designed to measure LLM performance on bioinformatics tasks.  The authors have curated 10 practical bioinformatics tasks with careful consideration to supporting evaluations of open source models and restricted running time (<4h) and memory requirements (<48GB).  Models were evaluated for planning quality and completion rate. Experiments were conducted to measure the variability of the agent outputs, and to determine the sensitivity of the agents to different perturbations by adding decoy inputs, corrupted inputs, and over prompting.

**Compliance With Llm Reviewing Policy:**

Affirmed.

**Final Justification:**

I believe that this is work provides a valuable contribution to the bioinformatics community by presenting a practical benchmark for standard types of tasks.  The authors have addressed my questions and provided a good justification for the scope of the work, balancing limitations on experimental resources vs. exhaustively characterize performance on the benchmark.  Given the aim of the benchmark is to evaluate bioinformatics-specific pipelines in realistic settings where robustness and workflow planning are critical requirements, the target audience of this benchmark should be practitioners who are already familiar with the tasks.  After some consideration, I've decided to maintain the weak reject recommendation. My view is that this work has more to offer to a specialized bioinformatics audience e.g. ISMB instead of ICML.  My chief concern is that an ICML audience may not be able to generalize the contributions of the work.

**Key Questions For Authors:**

Q1 What were the number of trials used to generate the % completion in Figure2?  I may have missed it
Q2 I was wondering what human level performance should be expected on BioAgent Bench, is it 100%?
Q3 Why was GPT-5.2 selected for the robustness testing?  The robustness results on the evolution task would be confounded by its low completion rate on that task.

**Limitations:**

yes

**Strengths And Weaknesses:**

Overall the paper identifies and aims to fill an important niche in benchmarking bioinformatics analysis pipelines.  The experiments are sound, though I felt there are some methodology gaps, such as a small number of replicates and relying on GPT-5.1 Lickert scale without human validation, likely leading to the scatter in Figure 3.   The selection of experiments for decoy data, corrupted data, and prompt bloat were nice to include and helped provide a window into the reliability of the models, though these tests were run only on GPT-5.2.  The work was presented very well, my only confusion was not carefully reading the Table 3 description where ✗=failure, ✓= pass for Corrupted inputs, but is reversed for Decoy inputs.  In terms of significance, the authors could have better contrasted BioAgentBench to other bioinformatics benchmarks like BixBench that seem to probe a superset of the tasks listed.  At the time of writing Opus-4.5 had already saturated the benchmark scores in Figure 2, and with the rate of open weight model improvement I expect saturation for that class of models soon.  GLM-5 may already be capable of this, limiting the value of the benchmark as a differentiator of ability.  The authors bring many good insights for evaluation and relevant tasks to bioinformatics, but with many other such resources for bioinformatics pipeline evaluation I feel this work should be further expanded and possibly merged with related benchmarks to provide a more comprehensive test of AI Agents.

---

> ### Author Rebuttal · Authors · 2026-03-28
>
> We thank the reviewer for the thoughtful and balanced review. We especially appreciate the recognition that the paper targets an important niche in benchmarking bioinformatics analysis pipelines, and that the perturbation experiments provide useful visibility into reliability beyond nominal task completion. We also appreciate the presentation feedback and the suggestion to better position the benchmark relative to related work. Below we address the specific questions and comments.
>
> **[SW]** We appreciate this perspective. Our intended contribution is not to replace broader biomedical or computational biology benchmarks, but to complement them with an evaluation suite centered specifically on end-to-end executable bioinformatics workflows requiring tool orchestration, file handling, artifact production, and robustness under controlled perturbations.
> This differs from benchmarks centered on exploratory analysis, broad scientific interpretation, or text response capabilities. In our view, the main value of BioAgent Bench is in providing a full system blueprint necessary for evaluating agents and models in this domain, ranking them and making visible the gap between "can finish" and "can finish reliably, robustly, and for the right reasons." Even when the completion rate becomes saturated, robustness, stability, and grounding remain meaningful differentiators, explored through the perturbation and robustness tests. We therefore view BioAgentBench as complementary to broader suites such as BixBench, where our evaluation suite can be adapted to other benchmarks and datasets to provide them with a targeted test of whether agents can complete realistic bioinformatics pipelines robustly and for the right reasons. We believe that tighter interoperability with broader benchmark ecosystems will be necessary in the future, but not a prerequisite for the paper’s present contribution.
> More broadly, we see the evaluation suite as useful not only for evaluating frontier models, but also for tracking progress in smaller or privacy preserving models, including distilled, finetuned, or systems with additional scaffolds (skills, MCP) that may be especially relevant in bioinformatics and clinical contexts where local deployment matters.
>
> **[Q1]** Figure 2 reports results from a single evaluation run per model. For each task, we record whether the agent successfully completed the task in that run, and we aggregate these outcomes into task level and overall completion rates. We chose this design deliberately, because our primary goal in Figure 2 is to measure end to end capability in a realistic single attempt setting, i.e., whether the agent can actually complete the workflow when deployed, rather than its best-of-N performance under repeated sampling.
> We treat robustness as a separate question, which is why repeated trials are analyzed in the robustness section rather than folded into the headline score.
> There is also a practical consideration, exhaustive multi-run evaluation across many models, tasks, perturbation conditions, and harnesses is financially and computationally expensive, particularly for frontier models whose bioinformatics trajectories can be long, tool intensive, with many artifacts generated along the way. That cost grows multiplicatively with the number of trials and evaluation settings, and we therefore chose single run completion as the primary benchmark metric, while using repeated trial analyses as a targeted diagnostic of reliability. We agree that larger scale multi run evaluation would strengthen the benchmark and statistical validity and view it as an important future extension.
>
> **[Q2]** We have not evaluated human bioinformaticians on the benchmark, so we do not want to speculate too strongly about a precise "human level" percentage. Designing such an evaluation would itself be a substantial undertaking, including decisions about whether humans are allowed to use AI assistance, whether one uses generalists or subfield experts, what time limits apply, and how many runs or retries are allowed. However this would be an interesting and valuable comparison.
>
> **[Q3]** GPT-5.2 was selected as a practical cost-performance tradeoff for running a larger number of robustness and perturbation experiments while still maintaining relatively strong completion performance. We agree with the reviewer that on tasks where the base completion rate is already low, perturbation-based degradation can be harder to interpret cleanly. That is a fair limitation of the current robustness setup.

---

> > ### Author Rebuttal · Reviewer_mcnT · 2026-04-02
> >
> > Thank you for addressing my questions, and clarifying the limitations and aims of the work.  In particular the emphasized aim to benchmark not just frontier models but any configuration of agents and system components.  One last minor follow up on Q2, I agree a comprehensive human benchmark would be interesting but a lower priority given the effort that would be required.  However it would be useful for a wide readership to provide a order-of-magnitude reference for the task difficulty, e.g. "we estimate a new bioinformatics Ph.D. would achieve a score of >80% with 2 hours of effort"

---

> > > ### Author Response · Authors · 2026-04-06
> > >
> > > Thank you for the follow-up. We agree that a human reference point would be useful for readers. Since we did not run a controlled human evaluation, we prefer not to provide a formal quantitative estimate without evidence. That said, because the benchmark tasks are grounded in standard bioinformatics workflows with human specified goals and expected artifacts, our expectation is that a trained bioinformatician with access to standard documentation and web resources would be able to complete all of the tasks successfully. The exact rate would depend mostly on the allocated time budget and further on subdomain expertise and what forms of assistance are allowed, which is why are avoiding making a claim in the paper without a controlled study.

---

### Official Review · Reviewer_o1j4 · 2026-03-11

**Soundness:** 3
**Presentation:** 2
**Significance:** 3
**Originality:** 3
**Overall Recommendation:** 4
**Confidence:** 4

**Summary:**

This paper presents BioAgent Bench, an evaluation suite for assessing the capabilities of AI agents in performing multi-step bioinformatics pipelines. Unlike existing benchmarks that focus on data analysis or isolated question-answering, BioAgent Bench frames tasks as end-to-end workflows requiring tool orchestration, file handling, and the production of structured outputs. The benchmark comprises of nine tasks across a few omics domains, and introduces robustness evaluations through input corruption, decoy files, and prompt bloat. The evaluation of state-of-the-art models reveals that while agents can successfully complete standard pipelines, they exhibit significant fragility under these perturbed conditions, often failing to validate inputs or ground their decisions in biological context. This work shifts the evaluation focus from task completion to reliability and appropriate failure detection.

**Compliance With Llm Reviewing Policy:**

Affirmed.

**Final Justification:**

After considering both the paper and the authors’ rebuttal, I remain at 4: Weak Accept. I think this is a useful and timely benchmark paper that addresses an important gap in evaluating AI agents on realistic bioinformatics workflows, and I found the focus on robustness and failure modes to be a real strength. My main concerns were about the evaluation methodology, especially the use of an LLM grader without validation against human judgement or evidence of consistency across different valid solution styles, as well as the limited scope of the robustness analysis and the necessarily simplified task setting. The rebuttal was helpful in clarifying the intended scope of the benchmark and in positioning the perturbation experiments as diagnostic rather than definitive, and I appreciated the authors’ candour about these limitations. That said, the central concern around the grader was not fully resolved: while I understand the argument that expert judgement is encoded upstream in the task design, this does not by itself establish that the grader is a reliable and well-calibrated measurement instrument for those expert-specified requirements. So overall, the rebuttal reinforced my positive view of the paper as a worthwhile contribution, but did not materially change my score, because I still see the lack of grader validation as an important limitation of the current benchmark.

**Key Questions For Authors:**

- You provide a strong rationale for using an LLM grader. Did you perform any human validation studies to confirm that the grader's scoring aligns with an expert's judgment? For instance, on a subset of trials, what was the agreement rate between the LLM grader and a human bioinformatician on metrics like steps_completed or results_match?
- Following from this, since you use an LLM grader to e.g. accommodate multiple valid solution paths, how do you ensure scoring consistency across trails that take fundamentally different approaches? For instance, could the LLM grader systematically favour more verbose trajectories, or those that structure their outputs in a particular way, even when the scientific validity is equivalent? Have you observed any systematic bias in the grader’s scoring across different methodological approaches to the same task.
- The paper identifies failure modes like "shallow file selection heuristics." Do you believe these are fundamental limitations of current agent architectures (e.g., their reliance on pattern matching), or could they be mitigated by better prompt engineering or a more sophisticated agent harness that forces a validation step before execution?

**Limitations:**

Yes.

**Strengths And Weaknesses:**

Strengths:
- The paper addresses an important gap in the evaluation of AI agents. As LLMs are increasingly being used as autonomous tools in scientific domains, benchmarks like this are essential for moving beyond ideal tests, and instead trying to understand their behaviour in messy, real-world scenarios. The focus on robustness is particularly timely and important.
- The benchmark is thoughtfully constructed with pragmatic constraints that enable reproducible, automated evaluation. By focusing on tasks that run under four hours with 48GB of RAM, the authors make it feasible to run large-scale agent evaluations without specialized infrastructure. The choice of tasks covers a representative slice of common bioinformatics workflows, and the experimental design, particularly the perturbation analysis, effectively supports the core message that completion rates overestimate reliability. This thoughtful construction, however, comes with an acknowledged trade-off (see Weaknesses below).
- The paper provides good insights into agent failure modes. These can directly inform the development of more robust agent scaffolds and prompting strategies.

Weaknesses:

- The authors make a compelling case for why an LLM grader is necessary. However, they do not provide a validation of the grader itself. It is unclear how well the LLM grader’s assessments correlate with human expert judgement. Without a small scale human eval, or an analysis of the grader’s reliability across different prompts or runs, it is difficult to know if the reported scores are accurate or if the grader is introducing its own biases and errors into the evaluation.
- By focusing on smaller organisms and well-packaged inputs to meet runtime and memory constraints, the benchmark excludes large-organism workflows (e.g., human sequencing) and omits common research tasks such as finding, downloading, and staging reference data. Consequently, BioAgent Bench cannot capture many of the most critical challenges in real-world bioinformatics, including messy metadata, massive genomes, and the need to source external references. This limits its ability to predict agent performance in high-impact clinical or research settings where these difficulties are central. The insights it provides are valuable, but they apply most directly to a simplified subset of bioinformatics work.
- While the robustness testing is a strength of this paper, its execution remains limited in scope. The perturbations are applied as a single trail per condition, offering only a qualitative snapshot rather than a statistically rigorous evaluation. A more comprehensive assessment would involve multiple runs to quantify variance and establish statistical significance across models. Additionally, many of the corruption methods are so severe that they would be trivially detectable even by simple scripted checks. The authors acknowledge that more insidious corruptions, which remain syntactically valid but are semantically or biologically implausible, represent the "safety-relevant" cases most critical for real-world deployment. However, these more subtle and realistic failure modes are not deeply explored in the current benchmark.

---

> ### Author Rebuttal · Authors · 2026-03-28
>
> We sincerely thank the reviewer for the thoughtful assessment of our submission. We especially appreciate the recognition that the benchmark addresses an important and timely gap, and that the robustness analysis helps reveal reliability issues beyond headline completion rates.
>
> **[W1]** The benchmark is not devoid of human expert judgment, rather the expert judgment is already encoded upstream in the task specification itself, the prompts (research goal) and the expected output structure and result formats that define successful completion. In that sense, the grader is not inventing evaluation targets from scratch, but assessing whether an agent’s trajectory and outputs satisfy expert specified requirements, without evaluating the more subjective layer of scientific or modeling choice among multiple biologically plausible alternatives, which is where expert review remains especially important. We view these roles as complementary. The benchmark is intended to evaluate execution in a scalable, software engineering setting, while human experts can remain in the loop for verification of biological interpretation and for steering agent behavior according to their own scientific standards.
> As noted in the paper’s limitations, LLM grading can be subjective and biased, and a human validation study would further strengthen the benchmark. We therefore view expert vs. grader validation, as well as more user or domain preference aligned grading, as important directions for future work.
>
> **[W2]** We agree with this point and view it primarily as a scope limitation rather than a contradiction of the benchmark’s goals. BioAgent Bench intentionally targets a tractable, reproducible subset of bioinformatics workflows, i.e. tasks that can be run repeatedly under bounded compute and memory, and that are structured enough to support automated evaluation. This design choice is what makes large-scale agent benchmarking feasible in the first place. Including large scale human data, external reference data retrieval, would substantially increase runtime, infrastructure demands, and evaluation complexity, and would have limited our ability to benchmark as many models, harnesses, and experimental settings.
> At the same time, we agree that this means the benchmark does not capture the full messiness of real world bioinformatics. Our robustness tests were intended as one step toward simulating this messiness in a controlled way, though we agree they cover only a subset of the possible real-world failure modes.
>
> **[W3]** We agree that the perturbation experiments should be interpreted as diagnostic rather than definitive. Their purpose in the current version is to surface concrete failure modes that are not visible from completion rates alone. We also agree that corruption severity matters, which is also how we have designed the decoy and corruption tests as explained in the paper, but we have not classified these perturbations with regard to their severity explicitly. Expanding perturbation coverage, adding subtler semantically invalid corruptions, and running repeated trials per condition are all important directions for future versions of the benchmark.
>
> **[Q1]** In the current version, we did not perform a human-validation study comparing the grader against expert bioinformaticians on a subset of trials. As discussed in W1, the benchmark already encodes human expert judgment. The grader operates over these expert specified requirements in a constrained way rather than attempting to replace expert judgment on scientific quality or biological validity. For these reasons, we chose a grading framework that can flexibly evaluate evidence of successful execution without requiring exhaustive manual review of all trials.
>
> **[Q2]** This is an important concern, and we agree it is a central challenge for agentic benchmark design. Our view, and experience, was that enforcing highly rigid scoring consistency, in this setting, over-penalizes legitimate variation in workflows, modeling choices, and intermediate representations. We therefore see this as a real tradeoff rather than something fully "solved" in the current paper and agree it merits deeper study in future work.
>
> **[Q3]** We believe this is one of the main takeaways of the paper: current agentic systems are often highly capable, but also surprisingly sloppy. Some of these failures can likely be mitigated through stronger prompting, better scaffolds, stricter validation policies, or tool-use constraints. But we also believe they reveal a broader weakness in current agent architectures, where raw capability does not reliably translate into careful and scientifically grounded execution.
> One motivation for BioAgent Bench is precisely to make these weaknesses measurable through the evaluation suite, which is useful both for comparing models and evaluating improved scaffolds, prompting strategies, and validation mechanisms.

---

> > ### Author Rebuttal · Reviewer_o1j4 · 2026-04-03
> >
> > Thank you for the thoughtful rebuttal. I appreciate the clarifications, especially on W2 and W3.
> >
> > My main remaining concern is still the evaluation methodology around the LLM grader. I understand the point that expert judgment is encoded upstream in the task design and expected outputs. However, this does not by itself establish that the grader is a reliable and unbiased measurement instrument for those expert-specified requirements. In particular, the rebuttal does not resolve whether the grader is stable across runs/prompts, or whether it may systematically favour certain trajectory styles or output formats over equally valid alternatives.
> >
> > I appreciate the authors acknowledging that this is not fully solved in the current paper. That honesty is helpful, and it keeps my view of the paper overall positive. However, since no validation or calibration evidence is provided, I still view this as an important limitation of the current benchmark.
> >
> > Overall, I still see the paper as a useful contribution, and I maintain my score.

---

> > > ### Author Response · Authors · 2026-04-07
> > >
> > > Thank you for your time, careful review and your feedback on the need for stronger validation of the LLM grader. We will take these points into account in our future work in the field.

---

### Decision · Program_Chairs · 2026-04-30

**Decision:**

Accept (regular)

**Comment:**

This paper describes BioAgent Bench, a benchmark for evaluating AI agents on end-to-end bioinformatics workflows. The benchmark also provides useful insights into agent failure modes that can inform future development of AI agents. Several concerns limit the strength of the contribution. The most significant issue is the reliance on an LLM-based grader without validation against human experts or analysis of consistency across runs and solution styles. Additional questions include limited statistical  evaluations (e.g., single trials per condition), simplified tasks relative to real-world bioinformatics, and relatively shallow analysis. The rebuttal clarifies design choices, but does not fully resolve concerns about evaluation methodology. Overall, the paper is a solid and useful contribution that is likely to be built upon, but further analysis would strengthen its impact.